# Detection of Diabetic Retinopathy Using Extracted 3D Features from OCT Images

**DOI:** 10.3390/s22207833

**Published:** 2022-10-15

**Authors:** Mahmoud Elgafi, Ahmed Sharafeldeen, Ahmed Elnakib, Ahmed Elgarayhi, Norah S. Alghamdi, Mohammed Sallah, Ayman El-Baz

**Affiliations:** 1Applied Mathematical Physics Research Group, Physics Department, Faculty of Science, Mansoura University, Mansoura 35516, Egypt; 2BioImaging Laboratory, Bioengineering Department, University of Louisville, Louisville, KY 40292, USA; 3Department of Computer Sciences, College of Computer and Information Sciences, Princess Nourah bint Abdulrahman University, Riyadh 11671, Saudi Arabia; 4Higher Institute of Engineering and Technology, New Damietta 34517, Egypt

**Keywords:** diabetic retinopathy, neural networks, thickness, OCT, reflectivity, classification

## Abstract

Diabetic retinopathy (DR) is a major health problem that can lead to vision loss if not treated early. In this study, a three-step system for DR detection utilizing optical coherence tomography (OCT) is presented. First, the proposed system segments the retinal layers from the input OCT images. Second, 3D features are extracted from each retinal layer that include the first-order reflectivity and the 3D thickness of the individual OCT layers. Finally, backpropagation neural networks are used to classify OCT images. Experimental studies on 188 cases confirm the advantages of the proposed system over related methods, achieving an accuracy of 96.81%, using the leave-one-subject-out (LOSO) cross-validation. These outcomes show the potential of the suggested method for DR detection using OCT images.

## 1. Introduction

Numerous conditions, including age-related macular degeneration (AMD), glaucoma, and diabetic retinopathy (DR) can affect the eyes [1,2]. These diseases range from visual fragility to vision loss, if not detected and treated early. Unfortunately, these diseases go unreported in the beginning and can only be identified by routine follow-up eye tests. This paper studies DR, a serious cumulative vascular and neurodegenerative condition that damages retinal cells without impairing vision yet is first difficult to diagnose. This disease is caused due to diabetes mellitus (DM), which is defined by poor glucose metabolism due to insulin failure or resistance. This can result in hyperglycemia, which can lead to vascular and neuropathic consequences. Thus, all diabetics are at risk of developing DR, especially in the adult population [3,4]. DR first manifests as a moderate condition with no obvious visual symptoms, but it can progress to a widespread and severe state, and the disease’s progression can result in blindness. Consequently, early diagnosis can lower the expense of therapy and the chance of sight loss to 57 percent [5].

According to Teo et al. [6], adults with DR are estimated as 103.12 million globally as of 2020. The number is anticipated to reach 160.50 million by 2045. On the other side, it was estimated that 28.54 million people could lose their vision owing to DR. The figures will rise to 44.82 million in 2045 [6]. Therefore, disease control is very required. It is necessary to start with the clinical care of DR, which entails enhancing the systemic control of blood sugar and blood pressure, where the management of blood sugar and blood pressure are the primary therapies. Although these treatments are very successful in diminishing the ratio of vision loss, they do not treat DR. There are three main treatments that are efficient in decreasing the vision loss ratio from this disease and are represented as follows: laser surgery, vitrectomy, and antivascular endothelial growth factor drug injection. Only the most severe instances of proliferative vitreoretinopathy are candidates for a vitrectomy, due to a dense vitreous tractional retinal detachment or hemorrhages [7,8]. Caution should be used when undergoing laser surgery since it harms retinal tissue [9]. Practically, those who seek therapy before their retina suffers substantial damage from advanced retinopathy have a chance of keeping their vision [10].

Specialized medical imaging of the eyes is a very important tool in early diagnosis. Fluorescein angiography, B-scan ultrasonography, fundus photography, optical coherence tomography (OCT), and OCT angiography (OCTA) are the most important specialized medical imaging tools for the eyes that are relied upon to assist in early diagnosis [11]. Noteworthy is the fact that the OCT device is of high quality, which makes any harmful changes easily noticed. In addition, it can be used to image the layers of the retina [12]. Therefore, this study uses OCT for the automated diagnosis of DR.

### 1.1. Related Work

On DR detection and diagnosis, several researchers have worked using retinal fundus photography images. For example, Priya et al. [13] used a model to diagnose DR using fundus images. This system was composed of four steps. First, preprocessing was done using an adaptive histogram equalization, a grayscale conversion, a matched filter, and a discrete wavelet transform. Second, fuzzy C-means clustering was applied for segmenting the blood vessels. From the segmented blood vessels, six features were extracted, which were the diameter, arc length, area, center angle, radius, and half area. Finally, three different classifiers were applied, which were a probabilistic neural network (PNN), a Bayes classifier, and a support vector machine (SVM). This system obtained an accuracy of 89.6% using PNN, 94.4% using the Bayes classifier, and 97.6% using the SVM classifier. In another study, Foeady et al. [14] used an automated system for DR diagnosis using fundus images. Preprocessing was applied to the images using histogram equalization, optical disk removal, contrast enhancement, filtering, and green channel extraction. Then, GLCM was used to extract statistical features such as contrast, correlation, energy, entropy, and homogeneity. Finally, they performed classification using an SVM. This system achieved an accuracy of 82.35%. Another study [15] applied a deep computer-aided diagnostic (CAD) system. First, preprocessing was applied to remove noise, improve quality, and resize the retinal images. Second, the authors extracted features, such as the gray-level co-occurrence matrix, regions of interest (ROIs), and the number of bifurcation points of the blood vessels. A multilabel SVM classifier was further applied to achieve an accuracy of 95.1%, specificity of 86.8%, and sensitivity of 86.1%. Rajput et al. [16] used fundus images to identify macular hard exudates, presenting an automatic approach based on mathematical morphology. By taking the extended minima transform’s complement, the hard exudates were extracted. If the scan revealed one or more hard exudates, it was then classified as diabetic maculopathy. Finally, the algorithm assigned the diabetic maculopathy a clinical significance level (CSME) or clinically nonsignificance level (Non-CSME) depending on how much the fovea was affected by the hard exudates. The accuracy, recall, and area under the curve (AUC) of the system were, respectively, 86.67%, 100%, and 97.06%. A related study [17] used a CAD method to grade diabetic maculopathy and DR. The accuracy of grading DR was 94.33%, while the specificity and sensitivity of diagnosing diabetic maculopathy were 98.56% and 96.46%, respectively. Rahim et al. [18] developed another system using six extracted features. These features were individually supplied to four different classifiers. The system achieved an accuracy of 70%, sensitivity of 45.28%, and specificity of 97.87% in the case of an SVM using a polynomial kernel. Using a naïve Bayes classifier, they achieved an accuracy, sensitivity, and specificity equal to 75%, 60.38%, and 91.49%, respectively. In the case of a K-nearest neighbors (KNN) classifier and an SVM using a radial basis function kernel, they achieved accuracies equal to 93%, 93%, sensitivities equal to 86.79%, 92.45%, and specificities equal to 100%, 93.62%, respectively. Sánchez et al. [19] employed a database of retinal pictures that was freely accessible to create a thorough DR CAD. They used 1200 digital color fundus pictures, and they extracted various characteristics to represent the structure, color, potential shapes, and contrast. These features were then used to feed a KNN classifier. This system had a sensitivity of 92.2%, a specificity of 50%, and an AUC of 87.6%. Mansour et al. [20] examined a CAD system on a Kaggle fundus dataset. They employed a multilevel optimization strategy that included preprocessing, region segmentation using a Gaussian mixture model based on adaptive learning, ROI localization using a connected component analysis (PCA), and feature extraction utilizing a PCA and linear discriminant analysis. Then, they fed these characteristics into an AlexNet-based categorization. With AlexNet, they achieved an accuracy of 97.93%. In another study, Li et al. [21] identified DR utilizing a transfer-learning Inception-v3 model, using 19,233 fundus images. Their method had a 93.49% categorization accuracy rate. Without using any pre- or postprocessing techniques, Abbas et al. [22] created a method for grading the five DR severity levels using 750 fundus pictures. Using a semisupervised multilayer deep learning method, their system obtained a 92.4% AUC, 92.18% sensitivity, and 94.50% specificity. Khalifa et al. [23] used a dataset to train and evaluate different deep learning models, such as ResNet18, VGG16, AlexNet, GoogleNet, and SqueezeNet. Their system achieved the highest testing accuracy of 97.9% using the AlexNet model. In another study, Kassani et al. [24] used an altered Xception architecture to classify the severity of DR. This method achieved 83.09% accuracy, 88.24% sensitivity, and 87.00% specificity. Rahhal et al. [25] used a neural network to categorize each picture into one of the five phases of DR using a publicly accessible dataset of fundus images. They achieved an accuracy of 66.68% for guessing the correct label for the image.

On the other hand, extensive studies have been done using OCT to identify various retinal disorders. For example, Ibrahim et al. [26] used a modified VGG16 CNN model, fused with ROI handcrafted extracted features, to diagnose drusenoid disorders, diabetic maculopathy, and choroidal neovascularization (CNV) using OCT imaging. Ghazal et al. [27] used a CNN system to analyze OCT B-scans for DR detection. They segmented the retina into 12 different layers. After that, they extracted patches from both the temporal and nasal sides of the fovea. Then, the patches, extracted from images were used for CNN training. Finally, an SVM was used for giving the final classification of DR or normal. This system achieved an accuracy of 94%. ElTanboly et al. [28] used a system to detect DR for patients with type 2 diabetes, using OCT images. First, they segmented the original OCT scan into 12 different layers. Second, they extracted features of the segmented layers such as curvature, thickness, and reflectivity. Finally, they performed a classification of each image whether diabetic or normal. They utilized a deep-fusion classification network with a stack of autoencoders that were not negatively restricted to classify the extracted features. This study achieved an accuracy of 92%, sensitivity of 83%, and specificity of 100%. Sandhu et al. [29] utilized OCT for DR grading using 80 patients. They segmented the retina into 12 layers and determined the thickness, reflectivity, and curvature features of each layer. Using these features, a deep learning network was trained to classify images as either normal or having nonproliferative DR. Their system obtained an accuracy of 95%. Li et al. [30] used OCT on 4168 images collected from 155 patients, to detect early stage DR. This study used a deep network, called OCTD-Net. This network achieved an accuracy, sensitivity, and specificity of 92%, 90%, and 95%, respectively. Aldahami et al. [31] utilized a system for DR diagnosis. They used 214 OCT images, where 107 images were normal and 107 were DR. Four statistical features were utilized: standard deviation and the matrix mean, median, and mode. The classification was performed using an SVM and a KNN classifier. Sharafeldeen et al. [32] used 2D morphology and reflectivity models for early DR detection using 260 OCT images.

Additionally, retinal disorders can be identified with OCTA. OCTA can provide views of the eye’s blood vessels and the retina. For example, Eladawi et al. [33] utilized a system for DR detection using OCTA images. This system was based on different preprocessing techniques, such as homogeneity, an initial estimation of a threshold, and a Markov random field model. They applied a segmentation step to extract the avascular zone of the fovea and vessels. Features were represented in vessel density, the width of the avascular zone of the fovea, and blood vessels’ caliber. The features were estimated to learn the SVM classifier. Finally, they achieved an accuracy of 94.3%, a sensitivity of 97.9%, and a specificity of 87%. Another study [34] used an ensemble deep learning technique using 463 OCTA volumes for DR diagnosis. Data were used to build neural networks, and ResNet50, DenseNet, and VGG19 architectures, pretrained on ImageNet, were used to fine-tune them. The network, trained on the VGG19 architecture, obtained the best accuracy of 92%, compared to the other architectures. Le et al. [35] used a CNN architecture, VGG16, to diagnose DR using OCTA. The method’s accuracy, specificity, and sensitivity were 87.27%, 90.82%, and 83.76%, respectively.

Other research combined OCT and OCTA modalities to identify the different grades of DR. For example, Sandhu et al. [36] used a system for grading NPDR using both OCTA and OCT on 111 patients. For OCT, they segmented the retina into layers. Then, they extracted features from each layer, which were represented in thickness, reflectivity, and curvature. For OCTA, they extracted the density of the vessel, the size of the avascular zone of the fovea, blood vessels’ caliber, and the count of crossover points and bifurcations. Finally, a random forest classifier was used for the classification. This system obtained an accuracy of 96%, a sensitivity of 100%, and a specificity of 94%.

### 1.2. Limitation of the Existing Works and Proposed Method

The main limitations of the previous works on OCT data are the following:Most of the techniques do not consider the 3D characteristics of the images and are limited to the 2D image features.A significant quantity of data is needed to train and test deep learning models, which may not be available.

To afford a better accuracy and make full use of the 3D characteristics of the retinal layers, this paper presents a novel 3D method for DR detection. The main features/contributions of the proposed method are the following:The paper is the first of its kind to analyze the 3D retinal layer by using low-level (first-order reflectivity) and high-level (3D thickness) information.Backpropagated neural networks are optimized to combine low-level and high-level information to further improve performance.Comparing the suggested approach to related methods, it performs better.

The remainder of this paper is structured as follows. The suggested methods are demonstrated in Section 2. Section 3 and Section 4 present the experimental results and related discussions. Section 5 concludes the paper and outlines its future directions.

## 2. Materials and Methods

### 2.1. Patient Data

Our developed CAD system was trained and tested on 3D-OCT volumes. Each OCT volume, which comprised five OCT image scans, was acquired for each eye (left and right). The data’s dimensions were equal to 1024 × 1024 × 5. This information was obtained using a Zeiss-Cirrus HD-OCT 5000 in the ophthalmology department of the University of Louisville (UofL) Hospital. The UofL Institutional Review Board (IRB) approved the acquisition and data collection. The study complied with the Helsinki Declaration. The dataset consisted of 188 cases (100 normal and 88 DR) with a minimum strength of signals of 7/10. Physicians with expertise in retinal diseases performed fundus exams to detect the disease.

### 2.2. Proposed Computer-Aided Diagnostic (CAD) System

The CAD system was separated into three steps, as illustrated in Figure 1. First, we used the 3D appearance-based segmentation approach [37] in order to segment the OCT images into 12 layers, as shown in Figure 2. Second, the features from each segmented layer were extracted, including the 3D first-order reflectivity feature in addition to the 3D thickness feature. In the third phase, the classification method was applied using the 3D features, extracted from each layer individually to obtain the final diagnosis. More details of the segmentation method, feature extraction, and classification technique are presented in the next sections.

#### 2.2.1. Segmentation of 3D-OCT Images

Using this method, the slice of the central area of the macula (fovea) was segmented into 12 layers. First, a Markov–Gibbs random field (MGRF) was used to segment the area from the retinal fovea. Then, a shape prior was formed that depended on OCT scans of the patients. From the shape prior, we could segment the neighboring slices. To get the final segmentation of a 3D-OCT scan, the prior shape models were repeated. Accordingly, the process of segmentation took place in several stages, which are summarized in the following steps:The B-scan OCT was aligned to a shape data formed by an expert. These shape data contained manual segmentations of the area in the center of the macula (foveal) of normal and diseased retina shape priors.The central B-scan was divided into twelve distinct layers based on intensity, MGRF spatial interactions, and shape.A nonrigid deformation B-scan was used as a prior shape pattern in each segmented B-scan process.The models of shape prior were repeated in each slice to get the final 3D segmentation.

An example of the segmentation of the OCT layers is shown in Figure 2. More details about the 3D segmentation approach can be found in [37].

#### 2.2.2. Feature Extraction

To describe the 3D segmented layers effectively, we extracted two features from each layer, namely, the first-order reflectivity and the 3D thickness.

The 3D first-order reflectivity was the first marker used in our approach. It expresses the light reflection intensity of a 3D individual layer. Because the reflectivity depends on some outer factors, it impacts on the quality of images, even in the condition where both eyes have no pathology and possess the same age, e.g., in the case of pupil dilation. Therefore, a normalization was applied on a given intensity of an OCT B-scan volume $\left(I_{i_{n}}\right)$ to control this problem as shown in Equation (Equation 1) [28].
(1)$$I_{i_{n}}=\frac{I_{i_{n}-RV}}{R_{1_{2-RV}}}$$
where *RV* is the mean intensity of the vitreous layer and $R_{1_{2}}$ is the mean intensity of the retinal pigment epithelium layer (RPE).

3D thickness was the second marker used to describe the OCT image in our approach. By computing the 3D Euclidean distance between each pair of related voxels on the layer boundaries, the 3D thickness feature determines the 3D thickness of a layer. To avoid intensity variations problems in the B-scans of retinal layers, we applied a geometric technique to measure the thickness instead of utilizing image intensities. This feature was evaluated by solving the 3D Laplace equation on the 3D segmented B-scan layers. The 3D Laplace equation is defined as in Equation (Equation 2):(2)$${▽}^{2}\gamma =\frac{\partial^{2}\gamma }{\partial x^{2}}+\frac{\partial^{2}\gamma }{\partial y^{2}}+\frac{\partial^{2}\gamma }{\partial z^{2}}=0$$
where γ(x,y,z) is the harmonic function or scalar field that depicts the estimated electric field between the layer’s target and reference boundaries.

The Laplace equation was applied to colocate the corresponding voxels. The second-order iterative Jacobi method was used to calculate γ(x,y,z) as in Equation (Equation 3).
(3)$$\gamma^{i+1}\left(x,y,z\right)=\frac{1}{6}\left\{\gamma^{i}\left(x+\Delta x,y,z\right)+\gamma^{i}\left(x-\Delta x,y,z\right)+\gamma^{i}\left(x,y+\Delta y,z\right)+\gamma^{i}\left(x,y-\Delta y,z\right)+\gamma^{i}\left(x,y,z+\Delta z\right)+\gamma^{i}\left(x,y,z-\Delta z\right)\right\}$$
where the predicted electric field at (x,y,z) during the *i*th iteration is denoted by $\gamma^{i}\left(x,y,z\right)$. Δx, Δy, and Δz are the step lengths in the directions of *x*, *y*, and *z*, respectively. Using the 3D Laplace equation, we summarize the steps of voxel-wise correspondences as follows:Locate the surface of the target and reference objects.Initially: adjust the minimum and maximum potential γ at the corresponding reference surface and the target surface, respectively.Using Equation (Equation 3), between both isosurfaces, γ can be estimated.Repeat the third step until convergence is achieved (i.e., no change occurs in the evaluated γ values between the iterations).

More details about the 3D Laplace equation can be found in [38].

#### 2.2.3. Classification System

After the extraction of the first-order reflectivity and 3D thickness, a classification system was applied based on two stages. First, two neural networks for each layer of the reflectivity and thickness features were built. Second, a fusing NN was used to fuse the first stage’s neural networks. The details of each network are as follows (see Figure 1):The first-order reflectivity feature of each layer was used to feed neural networks (NNs), each was composed of one hidden layer that involved 67 neurons. Each NN was applied to each layer individually. Through each NN, each layer gave a probability between 0 and 1The 3D thickness feature was fed to the NNs. Each NN was composed of one hidden layer containing 72 neurons and applied to each layer individually. The output of each NN was the probability of each layer.In the second stage, we fused the probabilities resulting from the previous first-stage NNs. The second-stage NN contained one hidden layer of 6 neurons. The output represented the final diagnosis.

The basic steps of NNs can be illustrated as follows:At first, the weights of NNs were initialized randomly.All outputs in hidden layers and output layers for neurons were calculated.The activation function was applied on each neuron of the outputs calculated in step 2.By using the backpropagation approach, the different weights were updated.Steps 2, 3, and 4 were replicated until the weights became stable.

#### 2.2.4. System Evaluation

We used different metrics to evaluate our system and determine its quantitative efficiency, where our system had two classes (the negative class represents normal cases and the positive class represents DR cases). These metrics were sensitivity, specificity, and accuracy [39]. Many experiments were conducted to validate our system, which were the 5-fold, 10-fold, and leave-one-subject-out (LOSO) experiments.

## 3. Results

To evaluate the quality of the proposed method, we compared it against other systems that depended on one extracted feature and against the most popular existing machine learning (ML) and deep learning (DL) methods. To optimize the NN architectures, their hyperparameters were adjusted by using various experiments. Those parameters involved the number of hidden layers (i.e., search space: 1–5) and how many neurons were present in each hidden layer (i.e., search space: 4–200). In the case of activation functions, we tried different functions for the output and hidden layers (range: softmax, sigmoid, RLEU, rectified linear, and tanh activation functions). In our system, the configurations were applied to get the best results represented in one hidden layer for all NNs, and the numbers of neurons in the hidden layers were 67, 72, and 6 for the reflectivity, thickness, and fusion NNs, respectively. In the case of activation functions, we applied the softmax function at the output layer and the tanh function at the hidden layers.

### 3.1. Results in the Case of Using the Reflectivity Only

As expected, the LOSO cross-validation achieved the highest accuracy compared to other cross-validation settings, since the number of training subjects was the maximum in this setting (see Table 1). As shown in the table, using the reflectivity extracted from one layer, the best layer accuracy was 82.45% using the LOSO cross-validation. The other settings (i.e., 5 folds and 10 folds) achieved best layer accuracies of 77.22% ± 8.65% and 79.23% ± 8.08%, respectively. When fusing the reflectivity feature from the 12 layers, the system achieved better accuracies (85.11%, 80.29% ± 7.66%, and 78.70% ± 7.48% using LOSO, 5-fold, and 10-fold cross-validations, respectively). These results indicated that the fusing of the reflectivity features from different layers improved the accuracy. However, the reflectivity feature alone was not able to provide the required performance.

### 3.2. Results in the Case of Using the Thickness Only

By the same analysis, using the thickness extracted from one layer, the best layer accuracy, using the LOSO cross-validation, was 89.36% (see Table 1). As illustrated in the table, the 5-fold cross-validation achieved an accuracy of 85.03% ± 7.84%, whereas the 10-fold cross-validation achieved an accuracy of 85.54% ± 10.16%. When fusing the thickness feature from the 12 layers, the system achieved an improved accuracy of 93.62% using the LOSO cross-validation, an accuracy of 91.40% ± 5.99% using the 5-fold cross-validation, and an accuracy of 90.34% ± 8.69% using the 10-fold cross-validation. These results indicated the importance of integrating the thickness of all layers to improve performance.

### 3.3. Results of the Proposed System

To integrate the differential effect of DR on each layer of the eye, the proposed system used a fusion neural network to fuse the layers’ features. This approach achieved improved results when compared to the other results presented (see Table 1). As demonstrated in the table, it achieved an accuracy of 96.81% using the LOSO cross-validation (the best setting accuracy, as expected), 94.12% ± 3.50% using the 5-fold cross-validation, and 94.67% ± 4.30% using the 10-fold cross-validation. These outcomes demonstrated the benefits of the suggested fusion system.

### 3.4. Comparison Results to Other ML Methods

To assess the advantages of the proposed fusion using NNs, we compared our proposed system with the most popular existing ML methods, such as SVM, KNN, binary decision tree (DT), and naive Bayes (NB) classifiers. The results of the comparison are illustrated in Table 2. As shown in Table 2, the proposed system achieved the highest accuracy. These outcomes demonstrated the benefits of the suggested neural-network-based fusion method.

### 3.5. Comparison Results to DL Methods

We compared the proposed system with the most popular DL algorithms, including Google Net and Resnet-50 (see Table 3). For these tested DL algorithms, we applied transfer learning to avoid overfitting. GoogleNet achieved the best DL accuracies of 93.98% ± 9.31%, using 5 folds and of 94.43% ± 9.45%, using 10 folds. By comparing the outcomes of the proposed system with the tested DL algorithms, it was clear that the proposed system was comparable with DL algorithms.

## 4. Discussion

One of the most serious diseases in the world is DR. Therefore, we proposed a CAD system that depended on retinal 3D-OCT image features to achieve an improved detection of DR. We used OCT images because OCT is one of the most common retinal examination techniques, which depicts the outstanding symptoms of DR. An OCT image shows the early symptoms of DR due to its ability to present retinal cross-sectional regions. The proposed framework depended on 3D-OCT images, which were segmented into 12 retinal layers using an automatic segmentation technique. The extracted retinal information from the segmentation process was collected and then passed to the proposed fused-NN model, which depended on the two 3D features extracted from the OCT scans. As illustrated in the result section, using each layer individually or using only one feature did not achieve a promising accuracy (see Table 1). One of the major aims of the proposed framework was to accurately diagnose DR. Thus, a fused-NN classification was proposed to integrate the information collected from all the layers and all the features. As reported in Table 2 and Table 3, the proposed framework obtained the highest diagnostic accuracy of 96.81%, compared to the most popular related ML and DL methods. The reasons behind the high accuracy of the proposed method were two folds: (i) using artificial intelligence (AI), especially the fusion NN, to fuse the reflectivity and thickness information, gathered from all the segmented layers, achieved an improved performance, and (ii) applying the method in 3D gave competing results, compared to other related works (see Table 4). Note that the comparison results of the related works in Table 4 were not fair, since all the compared methods were 2D techniques, whereas the data used in our study for evaluation were 3D (so they were not applicable to be evaluated on the same data). However, the results presented in Table 4 gave an insight into the advantages of using 3D-OCT data to improve diagnostic performance. To further investigate the advantages of the proposed 3D system, a comparison between using a 2D technique vs. a 3D technique on the same source data, using the same experimental setting, is shown in Table 5. As shown in Table 5, the proposed 3D technique on 3D data achieved an improved performance over the corresponding 2D technique on 2D data (using the slice on the center). This highlights the potential of using 3D information in order to obtain the fine details of the affected retinopathy parts in the OCT image, which improves the detection accuracy.

## 5. Conclusions

A new CAD technique to identify DR was used in this study to achieve protection against vision loss by using 3D-OCT. To test the proposed system, the study included DR and normal images. The proposed system segmented the 3D-OCT retina layers automatically. Then, it extracted two different features from the segmented retinal layers, which were the first-order reflectivity and 3D thickness. Furthermore, these two features were fused to train and test an NN classifier. The proposed system achieved promising results compared to other ML and DL algorithms. These results confirmed the advantages of the proposed method. In the future, we will try to test the proposed method on more data in order to investigate its robustness. In addition, other ML and DL algorithms will be investigated.

## Figures and Tables

**Figure 1 sensors-22-07833-f001:**
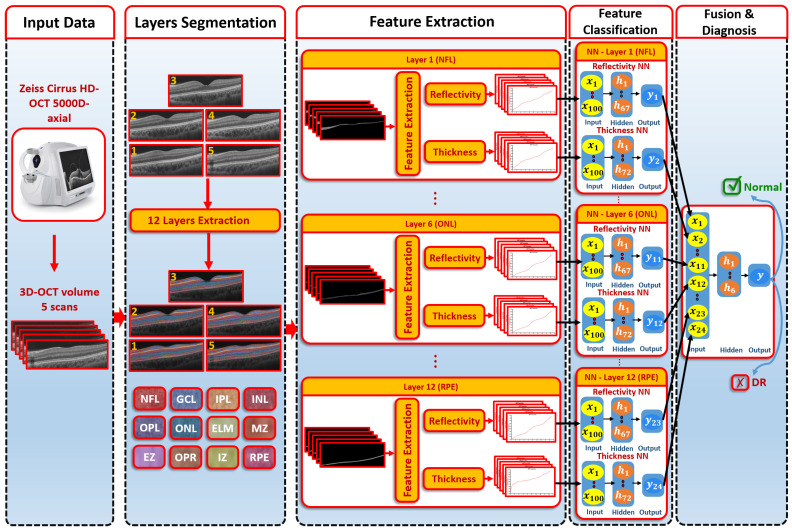
An illustration of the proposed 3D-OCT technology for DR diagnosis.

**Figure 2 sensors-22-07833-f002:**
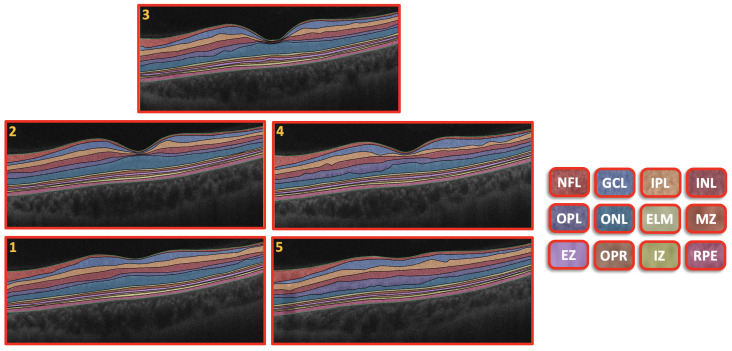
Retinal segmented layers starting from NFL layer and ending with RPE layer.

**Table 1 sensors-22-07833-t001:** Comparison results between the proposed system and other one-feature-based systems.

Model		Reflectivity			Thickness			Proposed System		
		Sens.	Spec.	Acc.	Sens.	Spec.	Acc.	Sens.	Spec.	Acc.
5 Folds	Layer 1	$59.80_{\;\pm \;13.03}\%$	$81.24_{\;\pm \;5.33}\%$	$70.29_{\;\pm \;7.46}\%$	$85.86_{\;\pm \;15.74}\%$	$83.39_{\;\pm \;8.49}\%$	$85.03_{\;\pm \;7.84}\%$	$93.23_{\;\pm \;6.57}\%$	$95.20_{\;\pm \;6.14}\%$	$94.12_{\;\pm \;3.50}\%$
	Layer 2	$69.88_{\;\pm \;14.99}\%$	$76.34_{\;\pm \;11.03}\%$	$72.98_{\;\pm \;10.53}\%$	$65.69_{\;\pm \;12.52}\%$	$78.89_{\;\pm \;13.83}\%$	$72.37_{\;\pm \;12.48}\%$			
	Layer 3	$74.15_{\;\pm \;14.69}\%$	$80.84_{\;\pm \;4.04}\%$	$77.22_{\;\pm \;8.65}\%$	$66.80_{\;\pm \;4.86}\%$	$73.74_{\;\pm \;15.97}\%$	$69.71_{\;\pm \;9.48}\%$			
	Layer 4	$79.45_{\;\pm \;22.48}\%$	$75.14_{\;\pm \;17.48}\%$	$76.11_{\;\pm \;10.24}\%$	$80.61_{\;\pm \;11.83}\%$	$68.59_{\;\pm \;14.27}\%$	$73.39_{\;\pm \;7.70}\%$			
	Layer 5	$69.31_{\;\pm \;17.74}\%$	$78.93_{\;\pm \;11.73}\%$	$73.48_{\;\pm \;10.00}\%$	$71.59_{\;\pm \;15.14}\%$	$48.22_{\;\pm \;23.81}\%$	$57.11_{\;\pm \;11.46}\%$			
	Layer 6	$68.66_{\;\pm \;11.14}\%$	$75.24_{\;\pm \;5.13}\%$	$71.26_{\;\pm \;4.81}\%$	$74.42_{\;\pm \;14.87}\%$	$54.88_{\;\pm \;13.74}\%$	$65.41_{\;\pm \;4.25}\%$			
	Layer 7	$67.40_{\;\pm \;10.64}\%$	$76.19_{\;\pm \;4.60}\%$	$71.35_{\;\pm \;5.95}\%$	$77.14_{\;\pm \;18.76}\%$	$45.81_{\;\pm \;20.03}\%$	$61.20_{\;\pm \;4.42}\%$			
	Layer 8	$65.95_{\;\pm \;8.48}\%$	$75.29_{\;\pm \;8.92}\%$	$70.18_{\;\pm \;6.20}\%$	$64.19_{\;\pm \;20.06}\%$	$59.63_{\;\pm \;16.46}\%$	$59.56_{\;\pm \;7.56}\%$			
	Layer 9	$69.66_{\;\pm \;17.88}\%$	$81.84_{\;\pm \;6.95}\%$	$75.50_{\;\pm \;9.48}\%$	$85.90_{\;\pm \;17.98}\%$	$47.36_{\;\pm \;32.50}\%$	$64.53_{\;\pm \;12.23}\%$			
	Layer 10	$46.26_{\;\pm \;5.49}\%$	$74.39_{\;\pm \;12.53}\%$	$61.08_{\;\pm \;8.77}\%$	$70.34_{\;\pm \;10.21}\%$	$68.74_{\;\pm \;20.18}\%$	$68.19_{\;\pm \;9.72}\%$			
	Layer 11	$63.73_{\;\pm \;11.11}\%$	$70.83_{\;\pm \;9.93}\%$	$67.05_{\;\pm \;7.87}\%$	$50.54_{\;\pm \;29.53}\%$	$68.28_{\;\pm \;18.09}\%$	$60.58_{\;\pm \;5.19}\%$			
	Layer 12	$55.38_{\;\pm \;16.85}\%$	$67.18_{\;\pm \;17.46}\%$	$60.06_{\;\pm \;5.94}\%$	$52.58_{\;\pm \;12.41}\%$	$77.64_{\;\pm \;11.84}\%$	$65.41_{\;\pm \;5.00}\%$			
	Fusion	$77.19_{\;\pm \;9.07}\%$	$83.80_{\;\pm \;7.68}\%$	$80.29_{\;\pm \;7.66}\%$	$88.37_{\;\pm \;12.01}\%$	$94.40_{\;\pm \;6.27}\%$	$91.40_{\;\pm \;5.99}\%$			
10 Folds	Layer 1	$62.16_{\;\pm \;14.18}\%$	$72.57_{\;\pm \;11.71}\%$	$66.93_{\;\pm \;8.51}\%$	$83.34_{\;\pm \;14.01}\%$	$88.68_{\;\pm \;9.53}\%$	$85.54_{\;\pm \;10.16}\%$	$97.46_{\;\pm \;5.50}\%$	$93.13_{\;\pm \;8.14}\%$	$94.67_{\;\pm \;4.30}\%$
	Layer 2	$76.77_{\;\pm \;21.51}\%$	$69.42_{\;\pm \;18.83}\%$	$68.51_{\;\pm \;9.17}\%$	$82.29_{\;\pm \;16.56}\%$	$67.52_{\;\pm \;32.83}\%$	$73.56_{\;\pm \;16.89}\%$			
	Layer 3	$79.22_{\;\pm \;17.12}\%$	$79.03_{\;\pm \;9.67}\%$	$77.71_{\;\pm \;9.09}\%$	$74.58_{\;\pm \;14.41}\%$	$74.51_{\;\pm \;22.48}\%$	$72.51_{\;\pm \;13.61}\%$			
	Layer 4	$75.67_{\;\pm \;20.17}\%$	$62.31_{\;\pm \;31.35}\%$	$65.02_{\;\pm \;17.02}\%$	$83.65_{\;\pm \;18.80}\%$	$71.76_{\;\pm \;18.65}\%$	$74.49_{\;\pm \;10.96}\%$			
	Layer 5	$77.50_{\;\pm \;21.76}\%$	$76.30_{\;\pm \;8.23}\%$	$74.49_{\;\pm \;12.29}\%$	$81.18_{\;\pm \;26.56}\%$	$36.14_{\;\pm \;36.43}\%$	$52.32_{\;\pm \;15.25}\%$			
	Layer 6	$80.37_{\;\pm \;21.05}\%$	$47.57_{\;\pm \;33.28}\%$	$63.44_{\;\pm \;11.42}\%$	$68.43_{\;\pm \;33.49}\%$	$40.97_{\;\pm \;35.38}\%$	$52.48_{\;\pm \;14.66}\%$			
	Layer 7	$72.02_{\;\pm \;24.82}\%$	$65.43_{\;\pm \;27.20}\%$	$68.70_{\;\pm \;12.07}\%$	$54.63_{\;\pm \;37.97}\%$	$61.13_{\;\pm \;34.78}\%$	$63.44_{\;\pm \;13.40}\%$			
	Layer 8	$85.21_{\;\pm \;17.58}\%$	$39.35_{\;\pm \;27.61}\%$	$59.38_{\;\pm \;10.71}\%$	$70.91_{\;\pm \;23.20}\%$	$61.41_{\;\pm \;12.16}\%$	$64.58_{\;\pm \;14.18}\%$			
	Layer 9	$76.75_{\;\pm \;16.16}\%$	$81.46_{\;\pm \;12.81}\%$	$79.23_{\;\pm \;8.08}\%$	$86.42_{\;\pm \;16.72}\%$	$44.64_{\;\pm \;18.83}\%$	$62.38_{\;\pm \;10.55}\%$			
	Layer 10	$55.50_{\;\pm \;25.53}\%$	$56.41_{\;\pm \;29.43}\%$	$55.88_{\;\pm \;12.72}\%$	$63.55_{\;\pm \;20.30}\%$	$67.05_{\;\pm \;17.73}\%$	$67.46_{\;\pm \;10.08}\%$			
	Layer 11	$62.78_{\;\pm \;17.70}\%$	$68.76_{\;\pm \;15.74}\%$	$67.00_{\;\pm \;10.19}\%$	$69.11_{\;\pm \;14.65}\%$	$41.15_{\;\pm \;16.18}\%$	$54.83_{\;\pm \;5.16}\%$			
	Layer 12	$53.34_{\;\pm \;20.24}\%$	$73.94_{\;\pm \;15.01}\%$	$64.43_{\;\pm \;10.87}\%$	$53.87_{\;\pm \;19.54}\%$	$68.32_{\;\pm \;24.34}\%$	$62.32_{\;\pm \;9.07}\%$			
	Fusion	$79.25_{\;\pm \;16.94}\%$	$79.26_{\;\pm \;12.33}\%$	$78.70_{\;\pm \;7.48}\%$	$88.86_{\;\pm \;14.63}\%$	$92.89_{\;\pm \;6.71}\%$	$90.34_{\;\pm \;8.69}\%$			
LOSO	Layer 1	77.27%	80.00%	78.72%	90.91%	88.00%	89.36%	95.45%	98.00%	96.81%
	Layer 2	82.95%	82.00%	82.45%	67.05%	83.00%	75.53%			
	Layer 3	84.09%	81.00%	82.45%	70.45%	78.00%	74.47%			
	Layer 4	81.82%	78.00%	79.79%	76.14%	75.00%	75.53%			
	Layer 5	81.82%	78.00%	79.79%	72.73%	43.00%	56.91%			
	Layer 6	71.59%	67.00%	69.15%	59.09%	49.00%	53.72%			
	Layer 7	61.36%	68.00%	64.89%	84.09%	44.00%	62.77%			
	Layer 8	82.95%	44.00%	62.23%	69.32%	59.00%	63.83%			
	Layer 9	77.27%	80.00%	78.72%	88.64%	39.00%	62.23%			
	Layer 10	62.50%	55.00%	58.51%	65.91%	82.00%	74.47%			
	Layer 11	53.41%	54.00%	53.72%	64.77%	55.00%	59.57%			
	Layer 12	50.00%	74.00%	62.77%	75.00%	70.00%	72.34%			
	Fusion	84.09%	86.00%	85.11%	92.05%	95.00%	93.62%			

**Table 2 sensors-22-07833-t002:** Comparison between other ML-based classifications and the proposed system.

Classifier		Sens.	Spec.	Acc.
5 Folds	SVM	79.07±12.62%	84.89±5.93%	81.46±7.29%
	KNN	81.17±12.38%	82.84±4.43%	81.43±6.57%
	DT	76.08±10.11%	78.84±11.31%	77.22±9.78%
	NB	65.98±12.16%	88.85±5.61%	77.75±7.37%
	Proposed System	93.23±6.57%	95.20±6.14%	94.12±3.50%
10 Folds	SVM	79.96±14.45%	84.50±8.85%	81.33±8.07%
	KNN	79.96±14.45%	84.50±8.85%	81.33±8.07%
	DT	72.65±11.67%	84.19±8.20%	78.17±6.80%
	NB	65.33±18.04%	88.85±7.38%	77.77±9.05%
	Proposed System	97.46±5.50%	93.13±8.14%	94.67±4.30%
LOSO	SVM	81.82%	89.00%	85.64%
	KNN	82.95%	89.00%	86.17%
	DT	75.00%	74.00%	74.47%
	NB	64.77%	90.00%	78.19%
	Proposed System	95.45%	98.00%	96.81%

**Table 3 sensors-22-07833-t003:** Comparison between DL classifiers and the proposed system.

Classifier		Sens.	Spec.	Acc.
5 Folds	Google Net	96.39±5.52%	92.00±13.04%	93.98±9.31%
	Resnet-50	97.50±5.59%	91.00±13.42%	93.98±9.68%
	Proposed System	93.23±6.57%	95.20±6.14%	94.12±3.50%
10 Folds	Google Net	96.67±7.50%	93.00±16.36%	94.43±9.45%
	Resnet-50	97.78±7.03%	90.00±18.86%	93.31±11.48%
	Proposed System	97.46±5.50%	93.13±8.14%	94.67±4.30%

**Table 4 sensors-22-07833-t004:** Reported related work accuracies using OCT data.

Related Work	Accuracy %
ElTanboly et al. [28], 2017	92.0
Sandhu et al. [40], 2018	94.3
Sandhu et al. [29], 2018	95.0
Li et al. [30], 2019	92.0
Ghazal et al. [27], 2020	94.0
Proposed System	**96.8**

**Table 5 sensors-22-07833-t005:** Comparison on 2D and 3D data using the proposed system.

Data Type		Sens.	Spec.	Acc.
5 Folds	2D	86.24±8.55%	82.59±6.68%	83.54±3.27%
	Proposed System (3D)	93.23±6.57%	95.20±6.14%	94.12±3.50%
10 Folds	2D	88.16±10.95%	84.54±13.61%	84.37±7.79%
	Proposed System (3D)	97.46±5.50%	93.13±8.14%	94.67±4.30%
LOSO	2D	93.18%	88.00%	90.43%
	Proposed System (3D)	95.45%	98.00%	96.81%

## Data Availability

Data are available upon reasonable request to the corresponding author.

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
