# Peer review of "Detection of Diabetic Retinopathy Using Extracted 3D Features from OCT Images"

_sensors, 2022, doi:10.3390/s22207833_

Round 1
Reviewer 1 Report
See attached!!!
I am glad to review the manuscript entitled Detection of diabetic retinopathy using extracted 3D features from OCT images. Although the overall novelty of the work is somewhat compromised, I still think it is may be accepted after some major problems are addressed to.
In this manuscript, the authors built a diagnostic pipeline based on 3D segmentation of OCT and feature extraction, and is capable of diagnosing DR on a private dataset. The segmentation method, as well as the feature extraction method, are previously developed by the authors.
Nowadays, the classification/diagnosis of OCT images is a heated topic in the area of deep learning, and many exciting new techniques have been developed. In this perspective, traditional methods may seem outdated.
However, I believe simple and effective systems are still important, and it may shed light on other related diagnostic tasks. This manuscript presents an amazing diagnostic capability, which is comparable with DL-based methods. Although it may partly be attributed to the relative simplicity of the task, it still demonstrates that traditional methods are powerful enough in some certain scenarios.
However, there are still some problems to be addressed to.
1. I noticed that an independent validation set was not partitioned. Please provide such results. Accordingly, you may want to explain your data preparation (OCT acquisition, train-test split, and image preprocessing) in the methods section, and remove lines 259–268.
2. Lines 168–169, I have never seen other people calling 2D/3D as ‘low level’ vs. ‘high level’. Please show me examples or please modify this.
3. In subsection 2.3, the classification system is not really clearly presented. Please consider using a figure to facilitate your explanation. You may also want to provide the details in adjusting the hyperparameters either in the results section or as a supplementary material.
4. Figure 3 is not necessary.
5. In subsection 3.5, it is more prudent to state that the proposed method is ‘comparable with’ DL algorithms. I would like to comment that the increase of accuracy alone does not really make a better system. So you may want to modify lines 307–309.
6. I understand that reproducing other related works is challenging, so I accepted the use of ‘reported’ values in comparison (Table 4). But I am wondering if it is possible to slightly modify your method to adapt to the 2D input? A comparison of 2D vs. 3D may reinforce your statement.
7. Please highlight the highest performance using bold typeface in the tables wherever appropriate.
Thank you!

Author Response
- I noticed that an independent validation set was not partitioned. Please provide such results. Accordingly, you may want to explain your data preparation (OCT acquisition, train-test split, and image preprocessing) in the methods section, and remove lines 259–268.
Response: We thank the reviewer for pointing this out. Lines 259-266 have been moved to the methods section. (Please see page 4, lines 177-185). Our data has been split using cross-validation. Namely, 5-folds, 10-folds, and leave-one-subject-out (LOSO). For example, 5-folds split data into five parts taking one part as a test and the rest as a train and repeating the process five times by taking a different fold each time as a test. Then, the performance average of these experiments has been calculated and reported in the manuscript. Our method involves no preprocessing steps. In the revised manuscript, OCT acquisition, and train-test split are discussed in the Methods section on page 4, lines 177-185, and page 7, lines 256-258, respectively.
- Lines 168–169, I have never seen other people calling 2D/3D as ‘low level’ vs. ‘high level’. Please show me examples or please modify this.
Response: We thank the reviewer for pointing this out. Low level / high level does not refer to 2D/3D. Low-level, which may be called Low-level features because it requires low-level processing of the original image, refers to the reflectivity information. High-level, which may be called High-level features because it requires high-level processing of the data, refers to the thickness information. In the revised manuscript, we have updated this sentence to remove any confusion. (Please see page 4, lines 168-169).
- In subsection 2.3, the classification system is not really clearly presented. Please consider using a figure to facilitate your explanation. You may also want to provide the details in adjusting the hyperparameters either in the results section or as a supplementary material.
Response: In the revised manuscript, the classification system has been illustrated in detail in Figure 1. (Please see page 5, Figure 1). also, we refer to this figure in the classification section (section 2.2.3). (Please see page 7, Line 236). In the revised manuscript, the hyper-parameters tuning has been illustrated in detail in the result section. (Please see pages 7-8, Lines 262-271).
- Figure 3 is not necessary.
Response: Thanks a lot. We have deleted it in the revised version.
- In subsection 3.5, it is more prudent to state that the proposed method is ‘comparable with’ DL algorithms. I would like to comment that the increase of accuracy alone does not really make a better system. So you may want to modify lines 307–309.
Response: Thanks a lot for this comment, which improves the readability of the paper. We have modified it according to your suggestion. (Please see page 10, Lines 308-309).
- I understand that reproducing other related works is challenging, so I accepted the use of ‘reported’ values in comparison (Table 4). But I am wondering if it is possible to slightly modify your method to adapt to the 2D input? A comparison of 2D vs. 3D may reinforce your statement.
Response: We thank the reviewer for pointing this out. We have compared our proposed system using 2D and 3D OCT data. As expected, using 3D data achieves better performance, which highlights the potential of using 3D information in order to obtain the fine details of the affected retinopathy parts in the OCT image. (Please see Table. 5 and Lines 333-339, pages 11)
- Please highlight the highest performance using bold typeface in the tables wherever appropriate.
Response: We thank the reviewer for his feedback. We have highlighted the highest performance in all the Tables. (Please see Tables. 1, 2, 3, 4, and 5, pages 9-11)
Reviewer 2 Report
In this study, the authors proposed a method to detect DR using deep learning. The technique is appropriate to perform segmentation of the retina and then load the information of each layer into the CNN.
Based on the OCT images, I would expect this study to have utilized horizontal sections. In horizontal sections, the GCL appears only on the nasal side. The GCL is almost completely absent on the temporal side. To determine glaucoma, OCT is often performed in vertical section. If it is a vertical section, the GCL is depicted on both sides of the image. The model in this study predicts a weak GCL determination. If the authors have not trained a sufficient number of vertical section OCT images, it is better to mention the above as a limitation.
Generally, neural network is called as (deep) convolutional neural network (DCNN or CNN). Almost all, the losses in CNN model optimized by backpropagation. I recommend modify the technical term.
Nothing else to point out. I think it is a good job.
Author Response
We sincerely appreciate the valuable comments, suggestions, and feedback provided by the reviewer on our manuscript. We thank you, in advance, for your careful consideration of our point-by-point responses given below. The revised version has been updated carefully to address all comments raised. Our point-wise response is provided below and is also reflected in the manuscript with yellow highlights.
- Based on the OCT images, I would expect this study to have utilized horizontal sections. In horizontal sections, the GCL appears only on the nasal side. The GCL is almost completely absent on the temporal side. To determine glaucoma, OCT is often performed in vertical section. If it is a vertical section, the GCL is depicted on both sides of the image. The model in this study predicts a weak GCL determination. If the authors have not trained a sufficient number of vertical section OCT images, it is better to mention the above as a limitation.
Response: We thank the reviewer for his feedback. We have trained the system using a sufficient number of OCT slices (i.e., mid-five slices). Also, we have compared our proposed system using the mid-slice only (i.e., 2D) and mid-five slices. The results showed that the proposed system achieved an
accuracy better than the 2D technique. (Please see Page 11, Table. 5, and Lines 333-339)
- Generally, neural network is called as (deep) convolutional neural network (DCNN or CNN). Almost all, the losses in CNN model optimized by backpropagation. I recommend modify the technical term.
Response: Thanks a lot for pointing out this point. The proposed neural network does not contain any convolutional layers, so, we can not call it a CNN. However, as you mentioned, the losses in this neural network is optimized using backpropagation. We have clarified this in Section 2.2.3, page 7, line 251.
Round 2
Reviewer 1 Report
I think the authors replied to most of the concerns appropriately. Although an independent validate set is highly recommended, it might be acceptable to use cross validation instead.